# Parametric Test of the Sentinel 1A Persistent Scatterer- and Small Baseline Subset-Interferogram Synthetic Aperture Radar Processing Using the Stanford Method for Persistent Scatterers for Practical Landslide Monitoring

**Farid Nur Bahti [1], Chih-Chung Chung [2],***  **and Chun-Chen Lin [1]**

[1] Department of Civil Engineering, National Central University, Taoyuan 320, Taiwan; faridnurbahti01@gmail.com (F.N.B.); hzop453gf7@gmail.com (C.-C.L.)

[2] Department of Civil Engineering/Research Center for Hazard Mitigation and Prevention, National Central University, Taoyuan 320, Taiwan

* Correspondence: ccchung@ncu.edu.tw; Tel.: +886-3-422-7151 (ext. 34120); Fax: +886-3-425-2960

**Abstract:** The landslide monitoring method that uses the Sentinel 1A Interferogram Synthetic Aperture Radar (InSAR) through the Stanford Method for Persistent Scatterers (StaMPS) method is a complimentary but complex procedure without exact guidelines. Hence, this paper delivered a parametric test by examining the optimal settings of the Sentinel 1A Persistent Scatterer (PS)- and Small Baseline Subset (SBAS)-InSAR using the StaMPS compared to the Global Navigation Satellite Systems (GNSS) in landslide cases. This study first revealed parameters with the suggested values, such as amplitude dispersion used to describe amplitude stability, ranging from 0.47 to 0.48 for PS and equal to or more than 0.6 for SBAS in WuWanZai, Ali Mt. The study further examined the suggested values for other factors, including the following: unwrap grid size to re-estimate the size of the grid; unwrap gold n win as the Goldstein filtering window to reduce the noise; and unwrap time win as the smoothing window (in days) for estimating phase noise distributions between neighboring pixels. Furthermore, the study substantiated the recommended settings in the Woda and Shadong landslide cases with the GNSS, inferring that the SBAS has adequate feasibility in practical landslide monitoring.

**Keywords:** landslide; Sentinel 1A; persistent scatterer (PS)-InSAR; small baseline subset (SBAS); global navigation satellite systems (GNSS)

## 1. Introduction

As a critical issue in mountainous areas, a landslide commonly comprises rock and soil materials that have moved downward along a slope [1], resulting in geohazards severely affecting the natural environment and human nature. Therefore, many researchers have dedicated themselves to characterizing the mechanisms or failure modes of rock landslides, which approximately include the following types: circular, plane, wedge, toppling, and falling [1,2]. In comparison, the classification of soil slope failures has been based on their type of movement: falls, topples, rotational, translational, lateral spread, flows, and complex [3]. However, the failure modes need field monitoring for further characterization, which requires much effort in practice.

Typically, spatial-point-based techniques are utilized to monitor the surface displacement: among these are the extensometer [4] and Global Navigation Satellite System (GNSS) [5]. Additionally, the inclinometer has been applied to in-depth deformations [6,7]. However, these point-based techniques have a spatial limitation over large landslides. To address the issue, multi-point procedures, such as the network GNSS stations, area solution to overcome this restriction and have been proven to be more effective [8]. However, these

techniques have an increased cost [9]. Therefore, space-borne-based technology could be an alternative to deliver more comprehensive monitoring over a wide area.

In the past few decades, the number of satellites used for scientific research has increased and the accuracy of surface monitoring with space-borne-based technology has gradually improved [10,11]. The usage of satellite-derived techniques is also helpful in landslide-studying, starting from landslide mapping [12], landslide susceptibility [13–15], and landslide monitoring based on the Synthetic Aperture Radar (SAR) and Interferogram SAR (InSAR) techniques [16–19].

The SAR technique utilizes electromagnetic (EM) radiation to obtain images of the Earth's surface in order to retrieve information about the ground surface without physically touching the object [20]. The SAR implements EM radiation in the visible to near-infrared range to capture the image without interference from clouds, sunrise, sunset, or other natural distractions [21]. Generally, the SAR sensor is separated into X-Band (wavelength equals 3 cm), C-Band (~6 cm), L-Band (~24 cm), and P-Band (~65 cm). Notably, the L-band wavelength (Advanced Land Observing Satellite, ALOS, for example) can provide a more excellent penetration capability to vegetation areas than the X-band and C-band [22,23], but has a low accuracy.

An open-source SAR satellite, Sentinel 1A, was launched by the European Space Agency (ESA) in 2004 and is flying with a short revisit time (within 12 days). However, producing a landslide early warning system is challenging, especially over mountainous areas with dense vegetation, since the Sentinel 1A operates in the C-Band sensor,. This limitation of the SAR method is critical to improving measurement accuracy [24]. Thus, extended SAR interpretations were proposed, starting from Differential InSAR (DInSAR) [25], Permanent Scatterer InSAR (PS-InSAR) [26], and Small Baseline Subset (SBAS) [27].

Although the InSAR process involves various monitoring methods, as stated previously, it is complicated because there are no exact criteria for each procedure or ranges for each parameter's value during processing, particularly for the application in landslide monitoring. This generates numerous questions regarding the InSAR-based monitoring of landslides: "What is the rule for InSAR processing using Sentinel 1A images?"; and "Does it have the exact guidelines to generate a consistent result?". The initial response to address the obstacles of the PS- and SBAS-InSAR techniques with the Stanford Method for Persistent Scatterers (StaMPS) approach was provided in [2,28]. Refs. [2,28] utilized a valid case study from the Maoxian landslide that occurred in June 2017, which was initially examined in [3,29]. Nevertheless, there is a lack of further criteria or quantifiable values regarding the StaMPS parameters, particularly in the context of monitoring landslides across vegetated mountainous regions [30,31].

Hence, the primary objective of this study is to present the most appropriate parameter values for using the Sentinel-1 InSAR with StaMPS processing as a proposed configuration. This includes the utilization of the PS-InSAR and SBAS-InSAR techniques, which will be validated using the GNSS data in the specific situation of the WuWanZai area. Moreover, based on the evaluation of the parameters, this study aims to validate the proposed configurations with the landslide case in Woda, China, along the Jinsha River, using the Sentinel-1 and the integrated GNSS data from [4,32], and the Shadong landslide with GNSS and SBAS monitoring near the Jinsha River Bridge [33]. Finally, a general guideline for the use of the Sentinel-1 InSAR with StaMPS in the practice of landslide monitoring is suggested.

## 2. Study Area

The WuWanZai, Ali Mt., Taiwan, was the primary location for the parametric test in this research that examined the InSAR parameters with GNSS positioning. Moreover, Woda [32] and Shadong [33] landslide cases with GNSS landslide monitoring were re-analyzed to evaluate the proposed parametric suggestions.

### 2.1. WuWanZai, Ali Mt., as a Test Case

The first field study was in WuWanZai, Ali Mt., Chiayi County, Taiwan, as shown in Figure 1. This district is located about 30 km from the east of Chiayi City. Alishan highway is a vital transportation section to Ali Mt. The significant interval between each elevation in Figure 1 represents a steep slope, where the maximum and minimum elevations are approximately 900 m and 540 m above sea level. WuWanZai is located in the nearby western foothills, where faults and folds have existed. The primary formation of WuWanZai is the Changzhikeng formation, which is composed of sandstone. Furthermore, there are two main streams (south Duzuo and north Duzuo streams) that pass through the north and south of the WuWanZai area. WuWanZai had an historical landslide triggered by rainfall in the past decade, and has now a ditch caused by erosion.

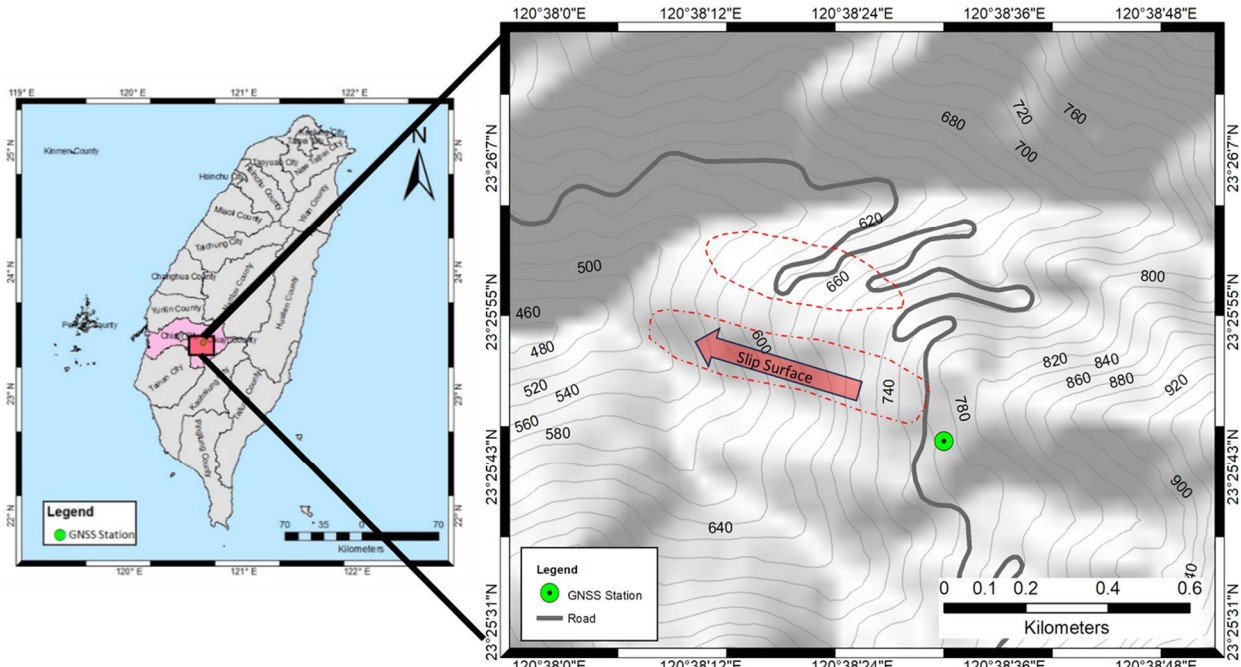

**Figure 1.** The location and topography map of WuWanZai (modified after the Central Geological Survey, 2009 [34]).

The profile of the WuWanZai landslide is illustrated in Figure 2, where the main landslide body is colluvium from borehole sampling. Although the comprehensive catchment wells were recently established to reduce the groundwater level and raise the stability of the WuWanZai slope, deeper sliding surfaces were considered as a deep-seated landslide, based on the previous boreholes and inclinometer results [35] and data provided by the Central Geological Survey [34], as depicted in Figure 2.

A GNSS station with u-blox F9P [36] as the continuous monitoring system was installed at the study location, as shown in Figure 1. The GNSS Receiver Independent Exchange Format (RINEX) data were retrieved automatically for Real-Time Kinematic (RTK) analysis in the NCU.

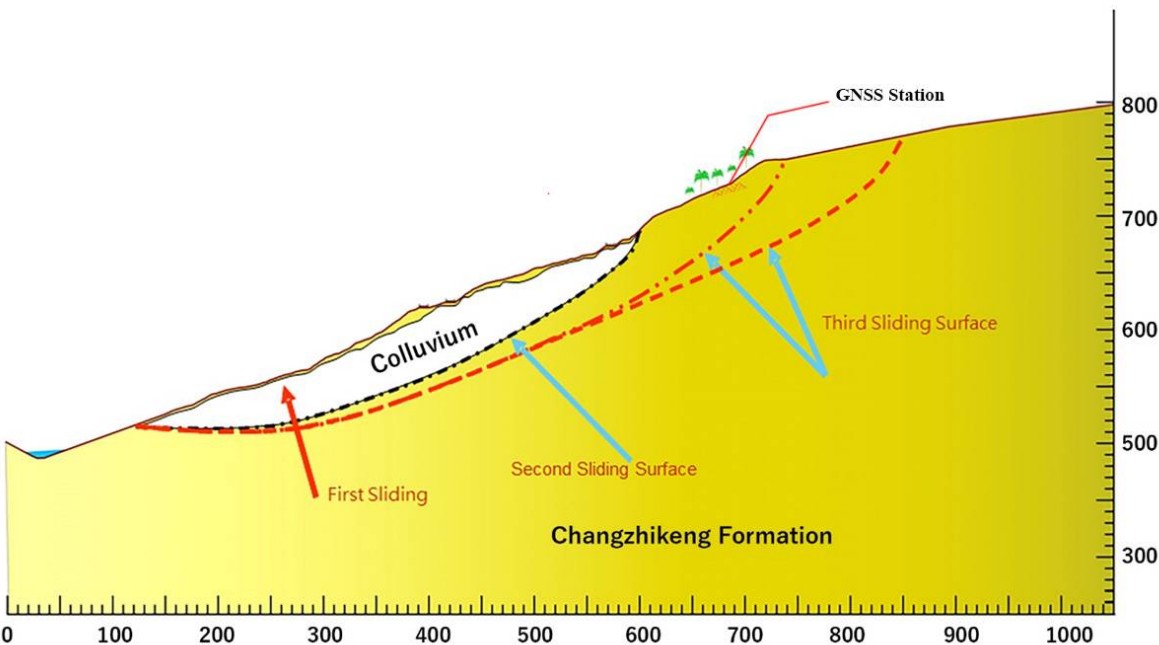

**Figure 2.** The slope profile of the WuWanZai (modified after [34,35]).

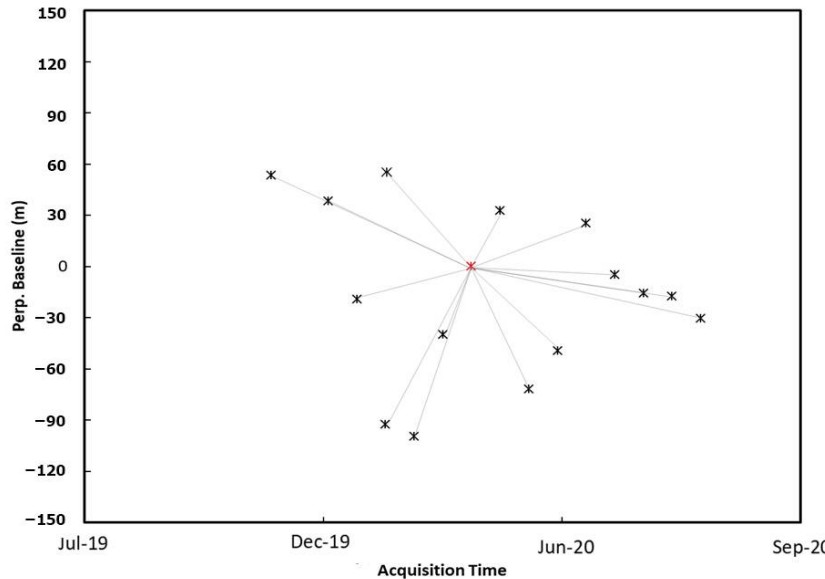

**Figure 3.** The baseline graph used in PS-InSAR.

Since the JHCI base station for RTK is located in the Zhugi township, approximately 13 km from WuWanZai, RTK was capable of processing the precise position (with the standard deviation being less than 10 mm) to continuously compare InSAR results too. As a preliminary test on the WuWanZai area, the Sentinel-A images from the 4 September 2019 to the 30 June 2020 were retrieved. The pictures from the descending satellite's direction were chosen because the WuWanZai slope faces west. Figures 3 and 4 show the baseline graph of PS- and SBAS-InSAR used in this case.

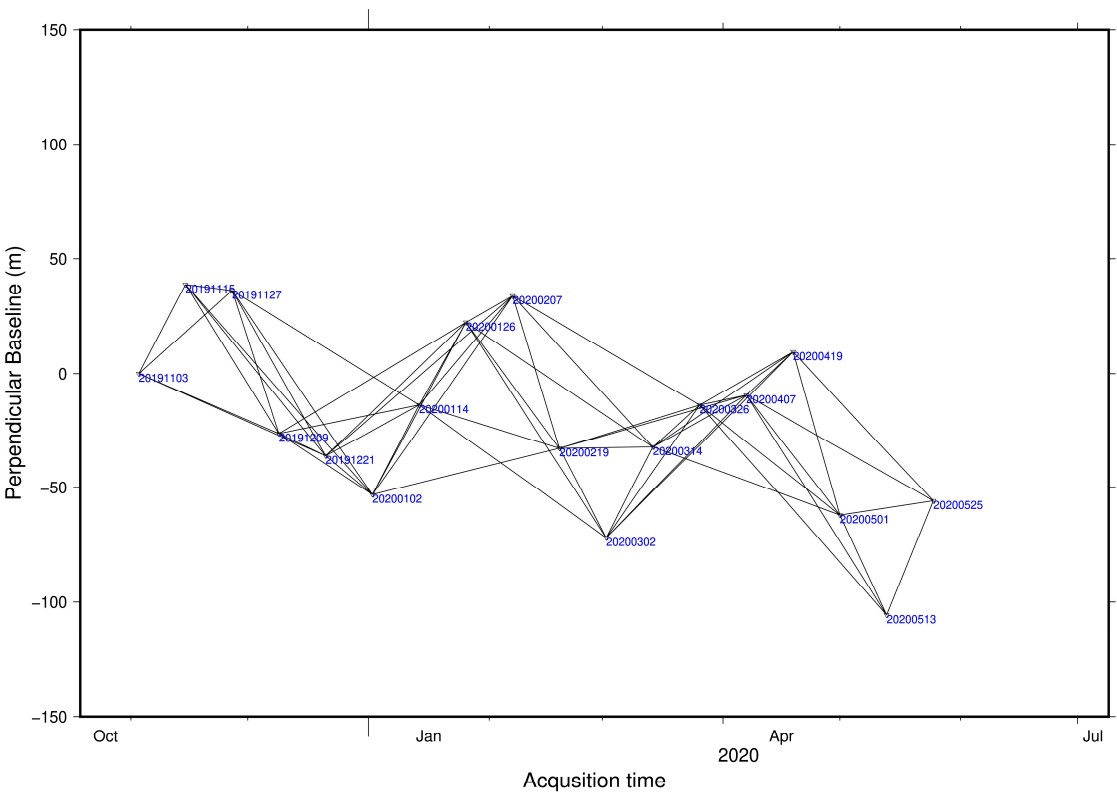

**Figure 4.** The baseline graph used in SBAS.

### 2.2. Woda Landslide as a Validation Case

As the validation case, the Woda landslide case was located in the Woda village close to the Jiansha River, Tibet, China [32]. The elevation difference on the site makes the location prone to failure. The surface elevation of the Jiancha River is approximately 2950 m, while the peak edge of the landslide location is around 4000 m, as shown in Figure 5. Hence, due to this significant elevation difference, Woda has a high risk of large-scale landslides. According to Li et al. [32], the geomorphological condition of the Woda area was typically a plateau alpine valley, where an inclination along the northwest and southwest Jiansha River was formed. Therefore, this terrain shaped a steep slope angle over the mountainous area. In addition, the terrain characteristic of the slope could be separated into three parts: top, middle, and bottom (see Figure 5). The top part had a steep slope angle with trees and vegetation. The central part was the gentle slope where the vegetation grew and forests scattered throughout the cultivated land. The bottom part consisted of cultivated land with growing trees. A GNSS station was set up in the bottom part, as depicted in Figure 5.

The descending images taken by Sentinel-A from July 2019 to August 2020, as revealed by Li et al. [32], were observed. All Sentinel-A images were applied in both PS- and SBAS-InSAR methods.

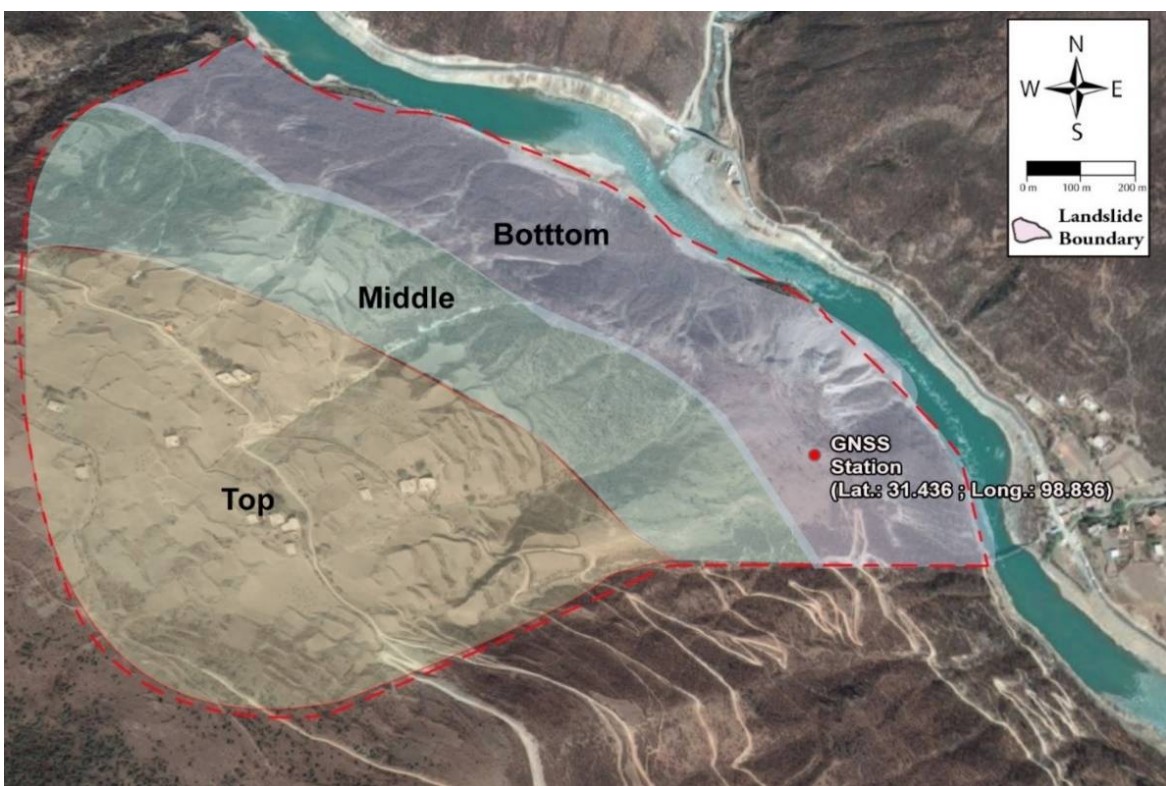

**Figure 5.** The Woda landslide location (modified after [4,32]).

### 2.3. Shadong Landslide Case

Multiple remote sensing technologies to monitor the landslide activity were combined to identify and observe large active landslides near the Jinsha River Bridge [33], as shown in Figure 6. SBAS-InSAR monitoring revealed a significant increase in the deformation rate of the Shadong landslide following the occurrence of an outburst flood, continuing until April 2021. The average deformation rate rose from 0.1 mm to 0.3 mm per day. Accordingly, a GNSS was employed to monitor the identified Shadong landslide, as depicted in Figure 6. Meanwhile, the Sentinel 1A's ascending images were used from the 22 December 2021 to the 3 May 2022 for the Shadong landslide, as revealed by Xu et al. (2023) [33].

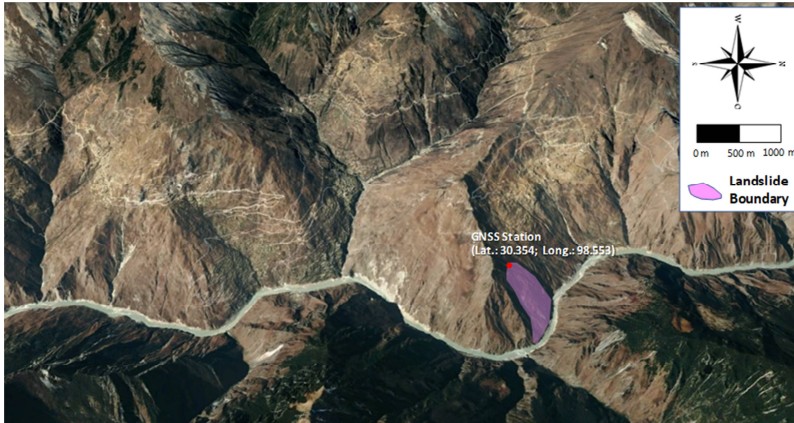

**Figure 6.** The Shadong landslide location (modified after Xu et al., 2023 [33]).

Utilizing the data gathered from these monitoring efforts, the Railway Survey and Design Institute adjusted the proposed bridge location to relocate it upstream of the active landslide cluster. This decision aimed to mitigate the potential danger of future landslide failures.

## 3. Materials and Methods

### 3.1. Processing Flow

The Persistent Scatterer Interferometric Synthetic Aperture Radar (PS-InSAR) and the SBAS-InSAR are popular techniques for studying surface deformations using radar waves. The Persistent Scatterer Interferometric Synthetic Aperture Radar (PS-InSAR) technique generates differential interferograms using a single master image to identify persistent reflectors. This method is most commonly used to investigate changes in urban areas with more constant reflectors than those in natural environments. In contrast to the more complicated SBAS technique, the PS-InSAR often uses a simpler deformation model, which is typically linear and does not require phase filtering. Hence, the whole working process is easier and faster than with the SBAS-InSAR.

On the other hand, the SBAS-InSAR technique depends on a network of image pairs, with a short spatial and moderate temporal baseline. This technology is able to identify temporal changes in surface deformations and enhance spatial coverage, particularly in non-urban regions. The SBAS approach uses the filtered and unwrapped phases to derive the deformation time series. This technique is more time-consuming from computational and operator-intervention standpoints because significantly more interferograms are created.

The InSAR workflow used two different methods, PS- and SBAS-InSAR. The SNAP software combined the StaMPS method [37] used for the PS-InSAR while using the StaMPS-SBAS method [38], integrated with the GMTSAR software adopted from Hayati et al. [39], to generate the SBAS result. Generally, both methods have a similar processing chain, which starts from obtaining the image data, parametric comparison, mapping, and validation using the GNSS data. As illustrated in Figures 7 and 8, the main difference between those methods was found in a pre-processing step, which will be explained later. Additionally, the reference points used in this study surround the GNSS station within a 100 m radius.

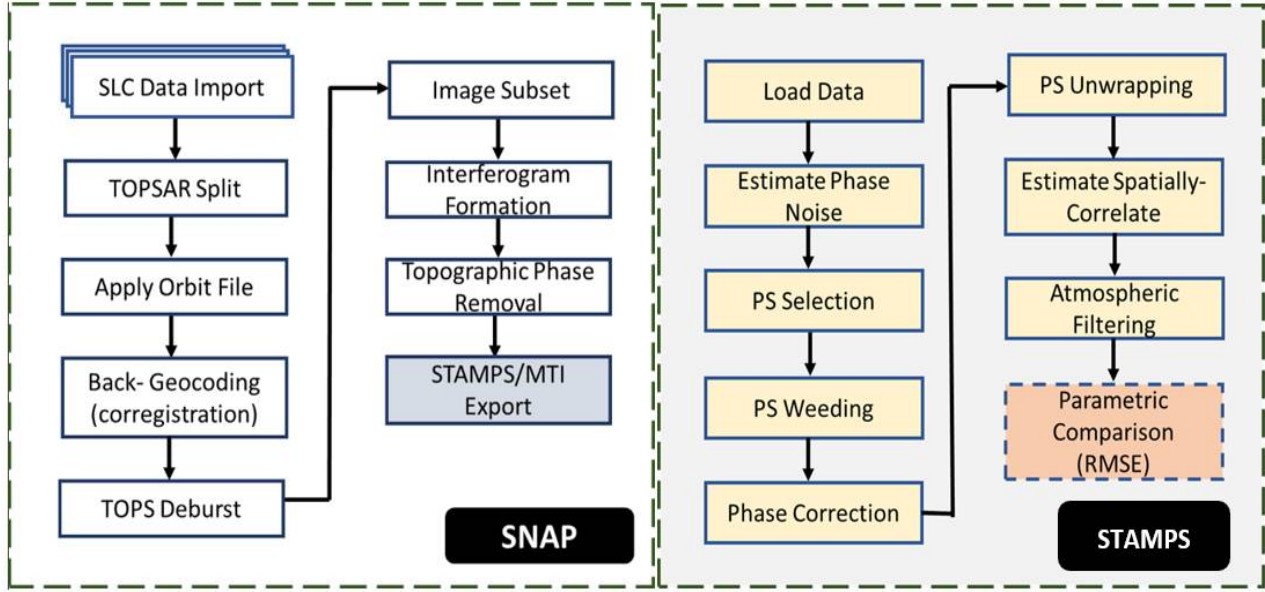

**Figure 7.** The General Workflows of the PS-InSAR analysis.

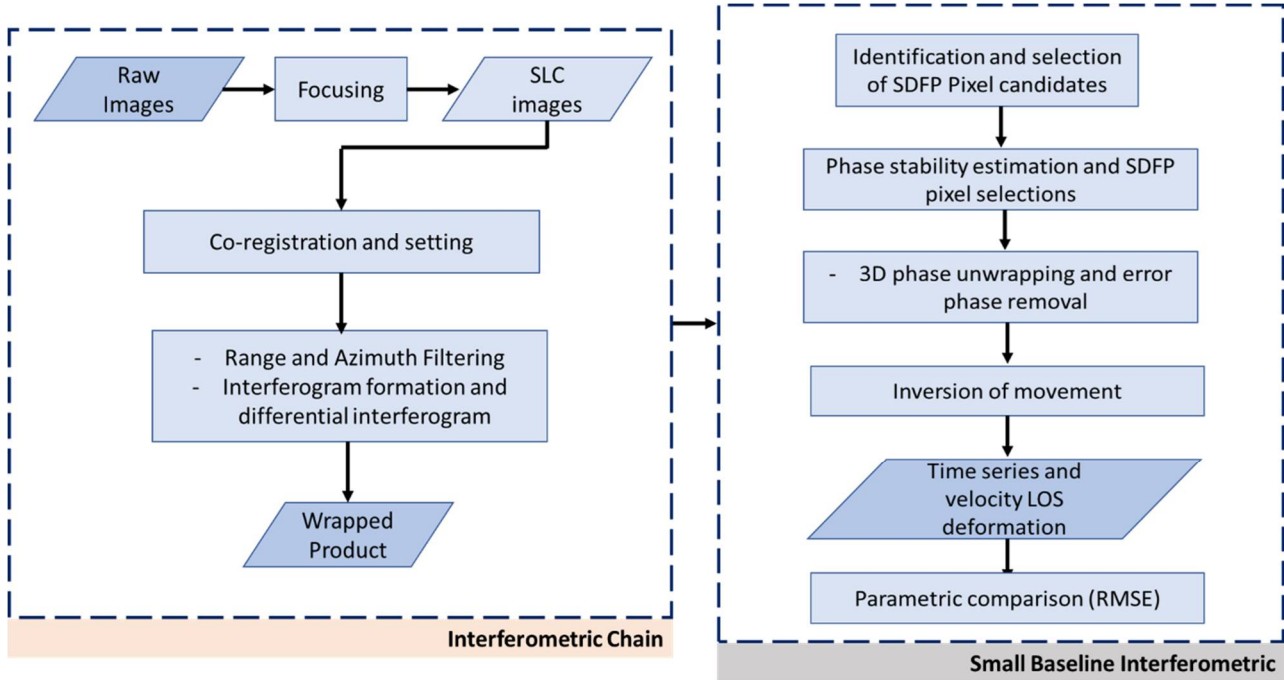

**Figure 8.** General workflow of the SBAS processing (modified after Hayati et al., 2020 [39]).

### 3.2. PS-InSAR Processing

Gabriel et al. [25] first invented the conventional DInSAR method. Then, the PS-InSAR was born in 1999 [26] to solve the limitations of temporal, geometrical decorrelation, and atmospheric inhomogeneities [40]. Since the growth of the PS method, Hooper et al. [37] have created a technique called StaMPS as an open-source software.

Two processing steps were applied in the PS-InSAR: a pre-processing step, with the SNAP software being used to export StaMPS version 4.1 format data; then, the study involved MATLAB® as the processor in the StaMPS step, and ArcGIS Pro® software was implemented to visualize the results. The proposed general workflow of the PS-InSAR is shown in Figure 7. For the preliminary study, in the StaMPS parameters, the following were selected for the test: *amplitude dispersion* were used to describe amplitude stability ranges, *unwrap_grid_size* was used to re-estimate the size of the grid; *unwrap_gold_n_win* was used as the Goldstein filtering window to reduce the noise; and *unwrap_time_win* was employed as the smoothing window (in days) that was used to estimate the phase noise distributions between neighboring pixels.

### 3.3. SBAS Processing

The SBAS-SAR technique introduced by Berardino et al. [27] was a way to overcome the limitations of the DInSAR. The SBAS-based detection and investigation were more applicable than the PS method in landslide monitoring [41,42]. The SBAS algorithm proposed by Hooper [38] was applied in this study, and the general workflow of the SBAS used in this study is shown in Figure 8.

The first step is the interferometric chain. This step consists in generating the interferograms using the GMT5SAR software [43] and converting the results to the StaMPS format. In this step, the interferogram formation determines the phase difference between two acquired images. After that, the adjustment related to the phase components will be estimated at the differential interferogram step. The topographic contribution will be reduced using available DEM images, and the measurement of the terrain motion component will be carried out during this step [44].

The second step is the small baseline interferometric step. The results of the GMTSAR software that have already been converted to the StaMPS format are then processed using the StaMPS method to generate the SBAS result.

The SBAS basic algorithm was similar to the PS-InSAR's, using the scatterer points. In addition, the SBAS utilized many master–slave pairs with small perpendicular baselines and short-time intervals, while the PS only applied a single master to generate the result. In addition, despite the PS and SBAS utilizing similar algorithms based on StaMPS, their main difference laid in generating interferogram pixel sets, where spatial filtering was not applied in a single master for the PS-InSAR. Instead, the SBAS method has the capability to perform three-dimensional phase unwrapping in both the spatial and temporal domains to retrieve the absolute phase retrieval for each pixel. Thus, the StaMPS parameters of *amplitude dispersion*, *unwrap_grid_size*, *unwrap_gold_n_win*, and *unwrap_time_win* were selected for testing.

### 3.4. Data Processing in StaMPS

### 3.4.1. Amplitude Dispersion

In the StaMPS method, persistent scatterer candidates (PSc) were selected by analyzing the amplitude that will affect the PSs. In the StaMPS approach, the PSs were estimated using the numerical simulation proposed by Ferretti et al. [26], which is as follows:

$$D_A = \frac{\sigma_A}{\mu_A} \tag{1}$$

where $D_A$ is the amplitude dispersion, and $\sigma_A$ and $\mu_A$ are the standard deviation and the mean of a series amplitude values, respectively. The default value of $D_A$ proposed by Hooper et al. [37] and Höser [28] is 0.4, which will be referenced in the PS-InSAR method as a single master image [45–47].

For the SBAS method, the amplitude dispersion was calculated using Equation (2) in order to obtain a better result in phase stability.

$$D_{\Delta A} = \frac{\sigma_{\Delta A}}{\mu_A} \tag{2}$$

where $\sigma_{\Delta A}$ is the standard deviation in the difference in amplitude between the master and slave images. The default value of $D_{\Delta A}$ proposed by Hooper et al. [38] is 0.6.

### 3.4.2. Phase Unwrapping

Initially, the InSAR phase was measured between $-\pi$ and $\pi$, which is known as a wrapped phase. Therefore, the unwrapping step was required in order to obtain the relative phase value [48]. In StaMPS, the three-dimensional unwrapping method was provided, consisting of two spatial scales and one time scale [37,46].

Each pixel's phase noise contribution was estimated before each unwrapped phase step was performed. The phase stability of each pixel was then examined from the estimation of the phase noise contribution with an iterative approach. The wrapped phase of pixel x (x-th) and interferogram (i-th) was given by the following Equation:

$$\psi_{x,i} = W\left\{ \phi_{D_{x,i}} + \phi_{A_{x,i}} + \Delta\phi_{S_{x,i}} + \Delta\phi_{\theta_{x,i}} + \Delta\phi_{N_{x,i}} \right\} \tag{3}$$

where $\psi_{x,i}$ is the wrapped phase, and $W\{\}$ is the wrapped operator. In terms of phase change, $\phi_{D_{x,i}}$ represents displacement along the line-of-sight (LOS) for each pixel; $\phi_{A_{x,i}}$ represents the phase contribution from atmospheric refraction; $\Delta\phi_{S_{x,i}}$ is the residual phase that is determined by the inaccuracy of the satellite orbit; $\Delta\phi_{\theta_{x,i}}$ is the look angle error caused by the residual phase; and $\Delta\phi_{N_{x,i}}$ is the different phase noise term.

Afterwards, the significant contribution of the spatial uncorrelated components was eliminated, such as the spatial correlation look angle ($\Delta\hat{\phi}^u_{\theta x,i}$) and spatial uncorrelated parts in "master" ($\hat{\phi}^{m,u}_x$). Thus, the terms in Equation (3) were subtracted and re-written as:

$$W\left\{\psi_{x,i} - \Delta\hat{\phi}^u_{\theta x,i} - \hat{\phi}^{m,u}_x\right\} = W\left\{\phi_{D_{x,i}} + \phi_{A_{x,i}} + \Delta\phi_{S_{x,i}} + \Delta\phi^c_{\theta x,i} + \Delta\phi_{N_{x,i}}\right\} \tag{4}$$

where $\Delta\phi^c_{\theta x,i}$ is the spatially correlated part of $\Delta\phi_{\theta x,i}$, and $\Delta\phi_{Nx,i}$ is the residual spatially uncorrelated noise term of $\Delta\phi_{N_{x,i}} - \hat{\phi}^{m,u}_x$. Then, the result of the three-dimensional unwrapping was formulated as follows:

$$\hat{\psi}_{x,i} = \phi_{D_{x,i}} + \phi_{A_{x,i}} + \Delta\phi_{S_{x,i}} + \Delta\phi^c_{\theta x,i} + \Delta\phi_{N_{x,i}} + 2k_{x,i}\pi \tag{5}$$

where $\hat{\psi}_{x,i}$ is the unwrapped value of $W\left\{\psi_{x,i} - \Delta\hat{\phi}^u_{\theta x,i} - \hat{\phi}^{m,u}_x\right\}$ and $k_{x,i}$ is the unknown integer for most *x* in each interferogram *i*.

In the unwrapping step, the observed phase value, known as modulo two rad, was used to estimate unambiguous phase values. Therefore, this was a critical process to interpret the actual displacement value.

*3.5. GNSS to LOS Projection*

The validation of the InSAR was assessed by comparing it to the GNSS data. The GNSS station in the first test was installed to validate the InSAR result. Since the radars could only measure path-length adjustments in the line-of-sight (LOS) direction, the GNSS positioning data needed to be transferred into the LOS direction. Based on Figure 7, if assuming the incidence angle and a satellite orbit with heading (azimuth) $\alpha$, the displacement $d_{LOS}$ in the LOS direction could be written as [49]:

$$d_{LOS} = d_u\cos\left(\theta_{inc}\right) - \sin\left(\theta_{inc}\right)\left[d_n\cos\left(\alpha_h - \frac{3\pi}{2}\right) + d_e\sin\left(\alpha_h - \frac{3\pi}{2}\right)\right] \tag{6}$$

where $d_u$ = displacement in vertical direction, $d_n$ = displacement in north direction, $d_e$ = displacement in east direction, $\alpha_h$ = satellite heading angle as shown in Figure 9a, and $\theta_{inc}$ = incidence angle as shown in Figure 9b.

The optimal values of critical parameters were examined by comparing the PS-InSAR and SBAS methods based on the GNSS data (daily data from averaging of second data). The PS- and SBAS-InSAR check-points surrounding the GNSS station within a 100 m radius were selected and averaged. Then, this study applied the root mean square error (RMSE) to the difference between the SAR and GNSS LOS variances as:

$$\text{RMSE} = \sqrt{\frac{\sum_{i=1}^n\left(I_i - \hat{G}_i\right)^2}{N}} \tag{7}$$

where *i* is the data index, *I* = PS-InSAR results, and $\hat{G}$ = GNSS linear regression, the blue line as illustrated in Figure 10. The individual GNSS displacement data were pre-processed to LOS projection based on Equation (6). Afterward, this study proposed the optimal values in each influential parameter by examining the lowest RMSE.

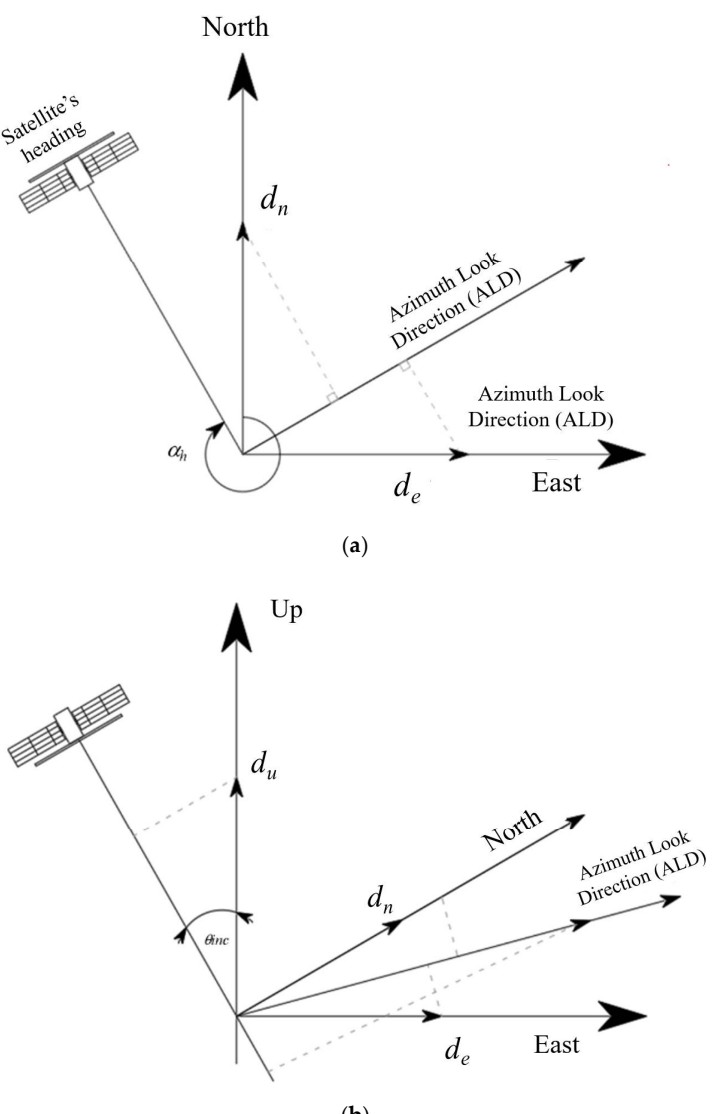

**Figure 9.** The line-of-sight velocity vector where the ascending satellite passes in the right-looking mode in (**a**) top view and (**b**) side view (modified after [10]).

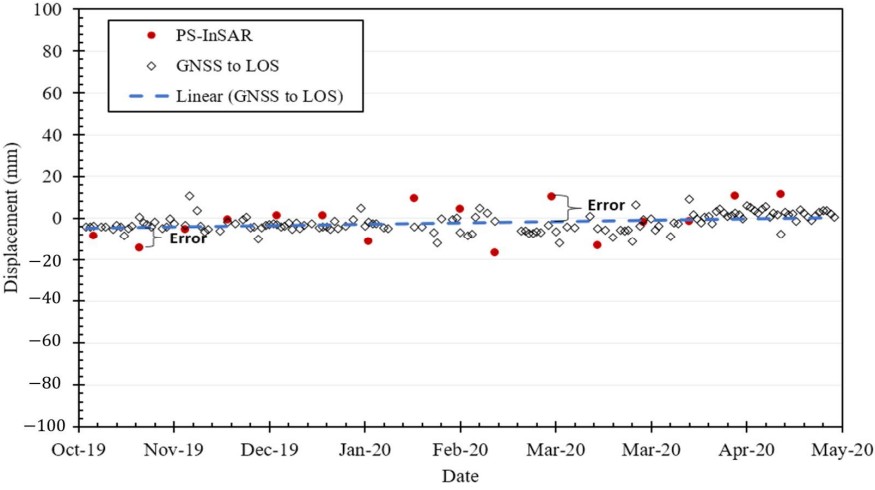

**Figure 10.** An example of the PS InSAR results compared to the GNSS in the RMSE model.

## 4. Results

### *4.1. The Preliminary Test Case at WuWanZai*

#### 4.1.1. Unwrap Grid Size

The *Unwrap Grid Size* is a control parameter to re-estimate grid size. The grid size will readjust after changing the *Unwrap_Grid_Size* value. Theoretically, grid size will affect increases or decreases in the SAR image noise, but, since the noise was increasing, the sum of signal sampling was lower. The default configuration was initially assigned a value of 200 m [50], but many sources in the literature used different values such as 10 m [28], 32 m [51], and 100 m [46].

In this study, after examining and comparing the GNSS data using the RMSE approach, the results, illustrated in Figure 11, show that the optimal values in this parameter for both PS-InSAR and SBAS are 20 m, and the value is suggested to be smaller than 50 m.

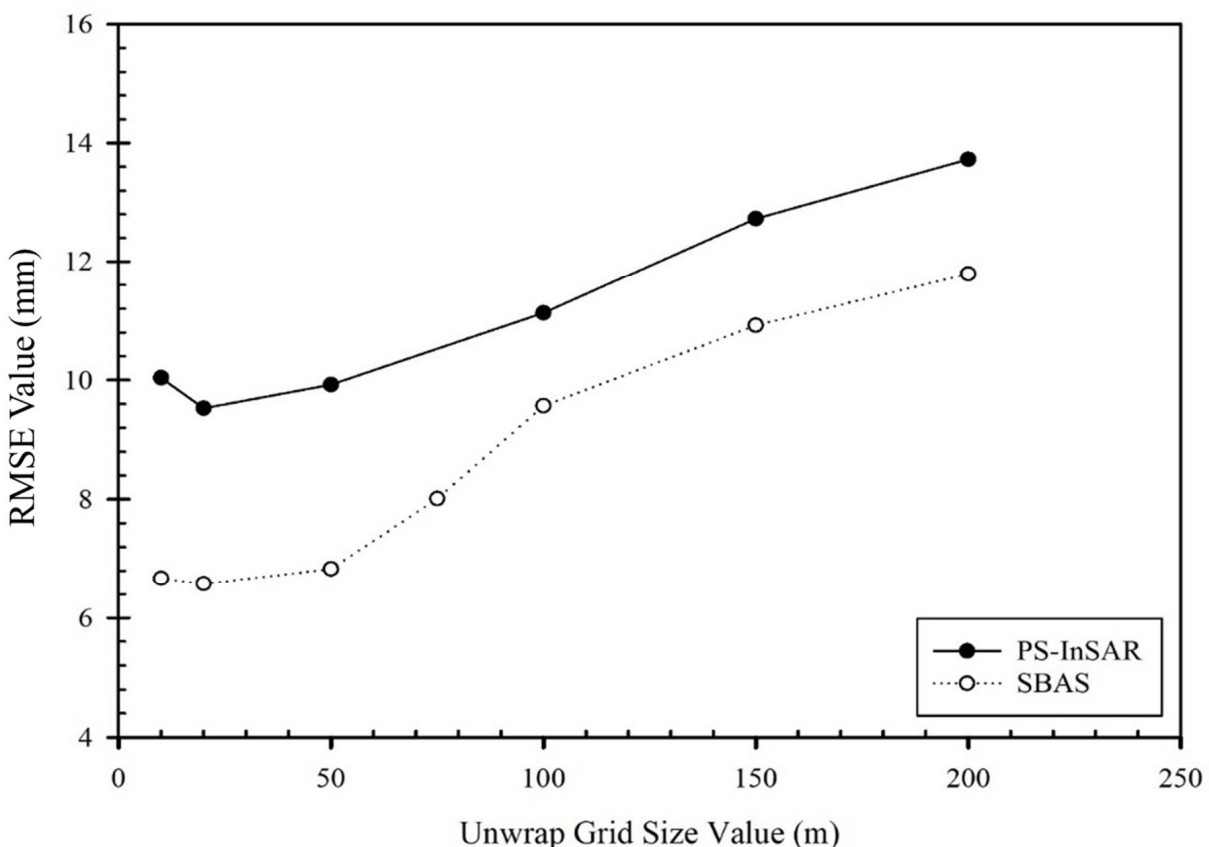

**Figure 11.** The comparison of the Unwrap_Grid_Size effect to the actual GNSS data at WuWanZai.

#### 4.1.2. Unwrap_Gold_n_Win

The *Unwrap_Gold_n_Win* is a parameter to reduce the noise using the Goldstein filtering method [52]. The preliminary test by [2,28] and Refs. [11,50] showed 32 as a reference. The comparison of the PS- and SBAS-InSAR results is shown in Figure 12. The result shows a turning point after about 32, and a significant change is noticeable after 24. Therefore, the optimal value of the *Unwrap_Gold_n_Win* parameter is ≤24.

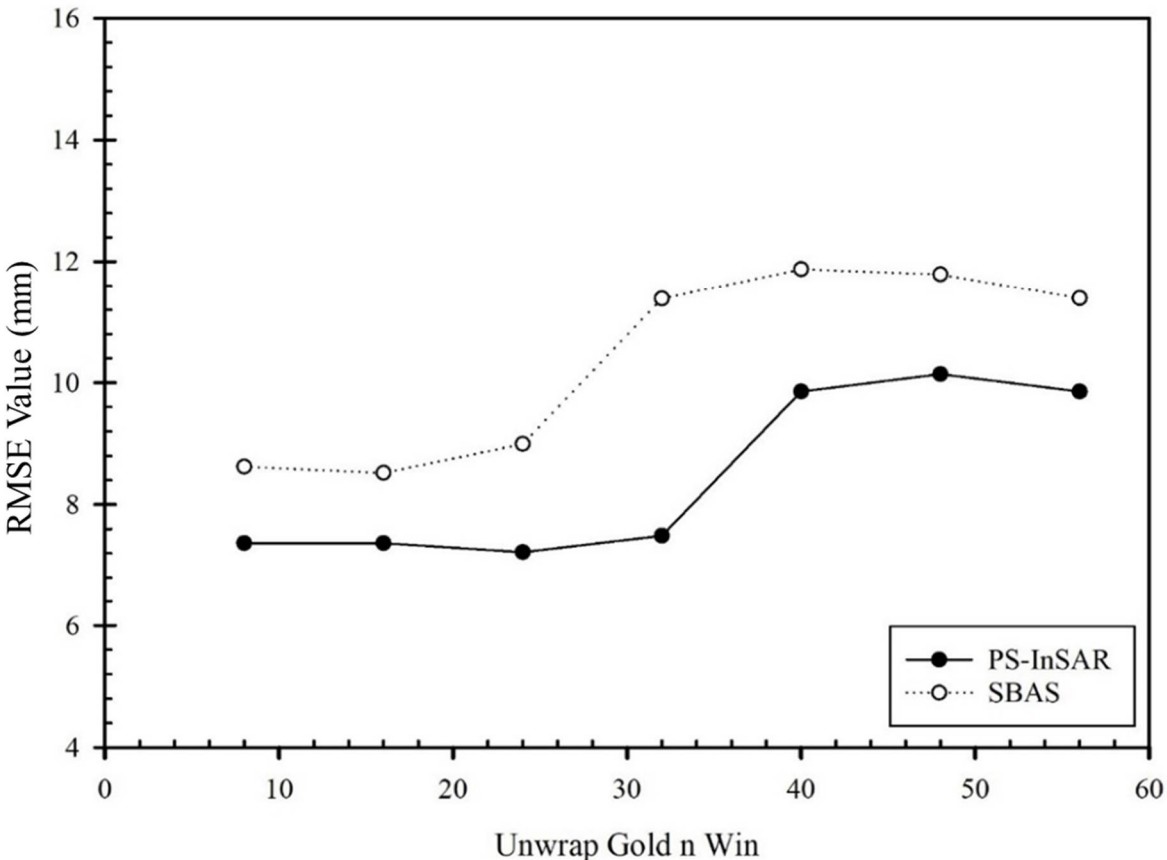

**Figure 12.** The comparison of the Unwrap_Gold_n_Win effect to the actual GNSS data at WuWanZai.

### 4.1.3. Unwrap_Time_Win

The smoothing window (in days) estimates phase noise distributions between neighboring pixels, such as a filter length in days, in which a phase is smoothed over time by calculating how noise contributes to phase arcs. The parameter *Unwrap_Time_Win* depends on how long the data are. Each pair of time series is smoothened using a Gaussian window. The previous results indicated the relevant values of the parameter *Unwrap_Time_Win* as 24 days [28], 30 days [1], and 730 days [1].

The values tested for the *Unwrap_Time_Win* in the PS-InSAR and SBAS methods ranged from 10 to 200 days, as shown in Figure 13. The turning point was depicted after 100 and 32 days for the PS-InSAR and SBAS, respectively. Hence, the suggestions are 100 days for the PS-InSAR and ≤32 days for SBAS. The suggested values are reasonable for the WuWanZai case.

### 4.1.4. Amplitude Dispersion

The PS InSAR method requires at least four PS candidates per $km^2$ over the spatial coverage area [53,54] to contain stable PS candidates [46]. The PSs are typically calculated after computing the dispersion index of the amplitude values for each pixel within the area of interest and considering only those targets that exhibit threshold values [26]. The amplitude dispersion index, DA, is applied to describe amplitude stability and reduce or increase the number of pixels in the phase analysis. The default value based on the StaMPS manual was 0.4–0.42, while the suggestion of Kampes [54] was <0.56, and Hooper et al. [46]'s was ≤0.6 to distinguish it from noise. Therefore, the range for the DA value was first set as $0.5 \leq DA \leq 0.3$ for the WuWanZai field case.

In the WuWanZai case, Figure 14 shows the distributions of the PS points with different DA values. The figure shows that it has fewer PSs when decreasing the DA value. Although

a more significant DA value induced numerous PSs, these PSs were getting unstable. Conclusively, the range for the DA value suggested was $0.47 \leq DA \leq 0.4$.

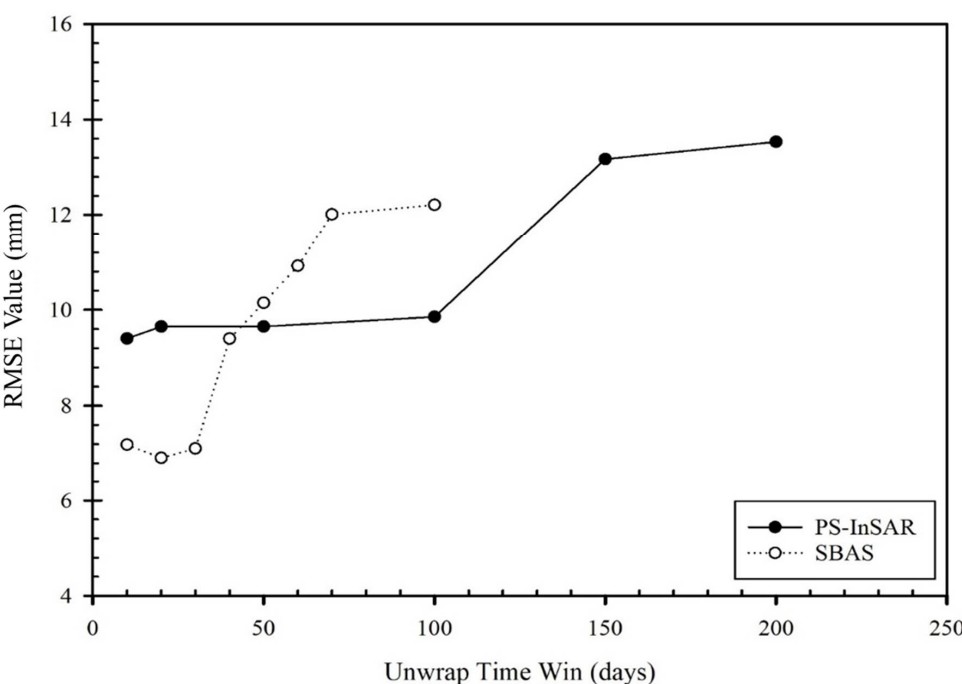

**Figure 13.** The comparison of the Unwrap_Time_Win effect to the actual GNSS data at WuWanZai.

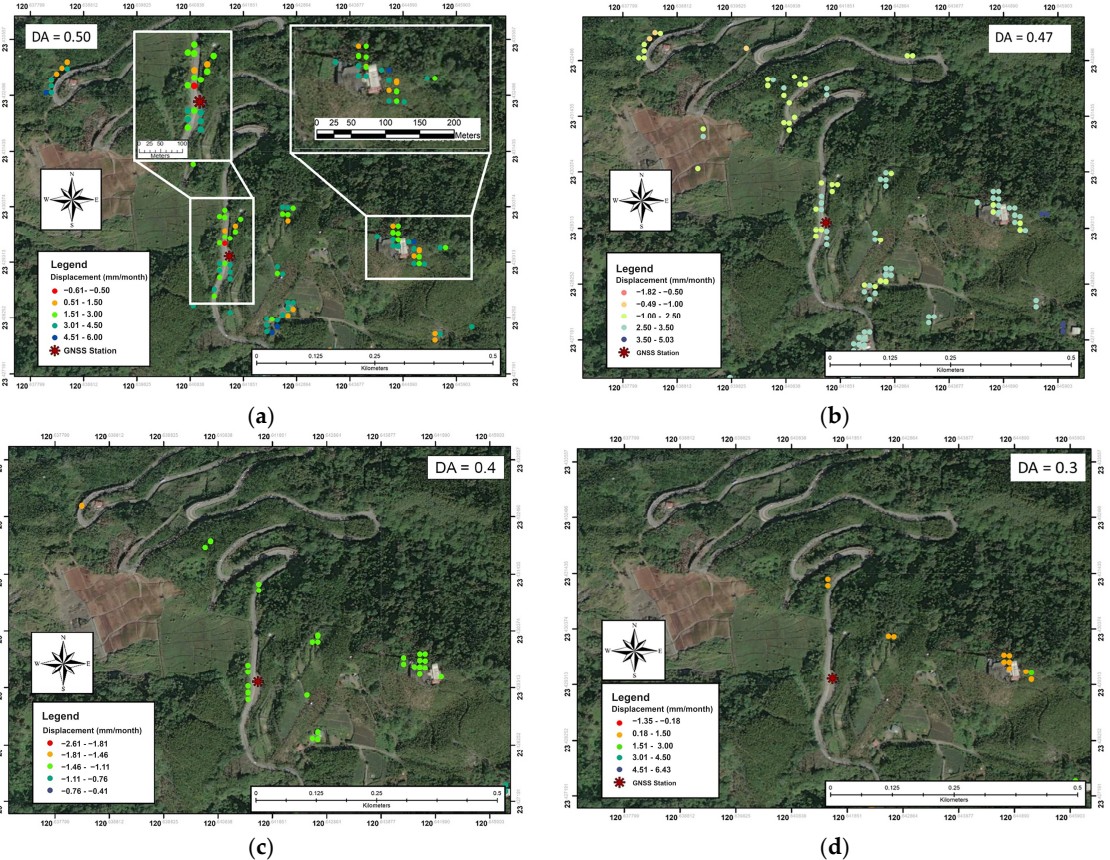

**Figure 14.** The distributions of the PS scatterer with (**a**) DA = 0.5, (**b**) DA = 0.47, (**c**) DA = 0.4, and (**d**) DA = 0.3, respectively.

Moreover, in the SBAS, the Gaussian scatter given by the DA applies the standard deviation among multiple masters and slaves. Figure 15a shows that the SBAS result has many scatterer points using DA = 0.6, while Figure 15b has fewer scatterer points of DA = 0.47, indicating the significant difference between both DA values.

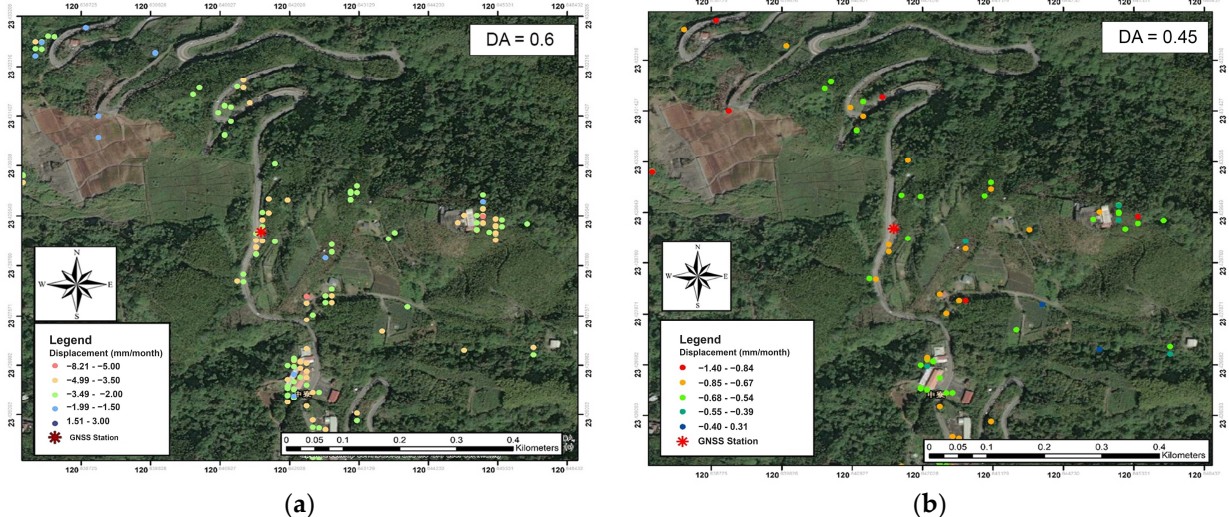

(**a**)                                                                  (**b**)

**Figure 15.** The distribution (DA) of the SB scatterer with (**a**) DA = 0.6 and (**b**) DA = 0.47, respectively.

The amplitude dispersion index obtained by Ferretti et al. [26] was theoretically equivalent. However, the SBAS has a better phase-stability estimation applied to spatial filtering [38]. The higher value is acceptable but will require extensive storage and time [38]. Thus, this study suggested 0.6 as the optimal value to reduce the computation time since processing the case with the DA value over 0.6 cannot be fully completed.

Based on Equations (2) and (3), the standard deviation of a series of amplitude values is considered in DA estimation. Although the quality of the PS/SBAS points is not considered, the results that were rigorously verified with the GNSS data may be useful in practice.

### 4.1.5. Summary of Test

The influential parameters and the corresponding optimal values of both PS- and SBAS-InSAR were examined in this study, as suggested in Table 1.

**Table 1.** The optimal values of influential parameters in PS- and SBAS-InSAR.

| Parameter | Default Value | PS | SBAS |
| --- | --- | --- | --- |
| Amplitude Dispersion (DA) | 0.6 | 0.4–0.47 | ≥0.6 |
| Unwrap_grid_size (m) | 200 | 20 | 20 |
| Unwrap_gold_n_win | 32 | ≤32 | ≤24 |
| Unwrap_time_win (days) | 730 | ≤100 | ≤32 |

### 4.2. Verified Case Results at Woda Landslide

The Woda landslide case was applied to validate the suggestions mentioned above. Li et al. [32] used the SBAS method to compare scatterer points with the GNSS data, Therefore, the similar model can be generated by using the suggested values and compare it to the previous result as shown in Figure 16. By implementing the PS- and SBAS-InSAR optimal values in the Woda case, this study found only a few PSs in the PS-InSAR result, resulting in the difficulty of GNSS comparison. This explains that the fewer Sentinel-1 images affected the reliability of the PS-InSAR results. Fortunately, the images of the SBAS-InSAR were sufficient for the analysis in this case. Thus, Figure 16 only shows the data of the GNSS in the LOS direction and the SBAS displacements after the optimization.

The RMSE values indicate significant improvement compared to the default values of Li et al. [32].

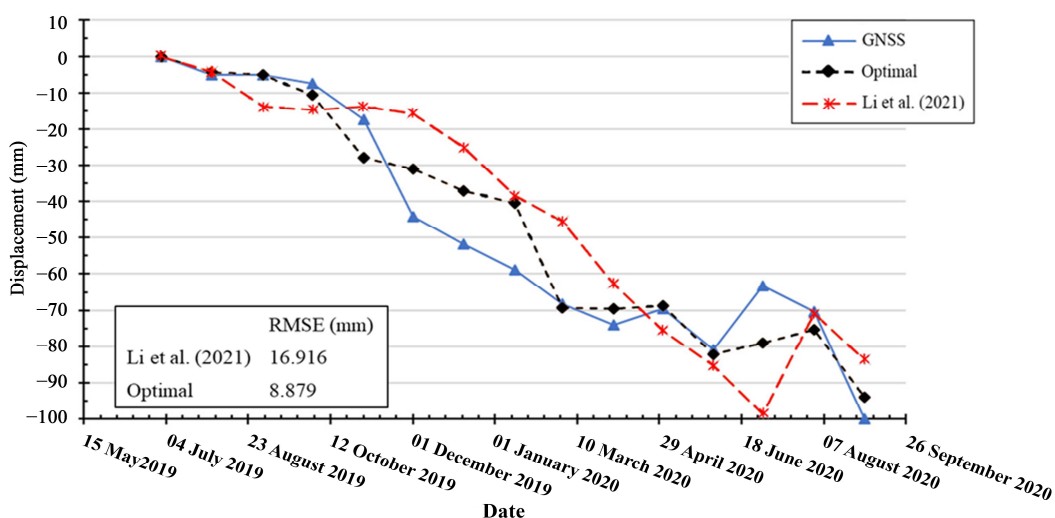

**Figure 16.** The comparison of the optimal SBAS and Li et al. (2021) [32] results with GNSS.

### 4.3. Verified Case Results at Shadong Landslide

The Shadong landslide case was further applied to validate the suggestions mentioned since Xu et al. [33] revealed the SBAS method to compare with the GNSS data. Similarly, by implementing the PS- and SBAS-InSAR optimal values in the Shadong case, this study found only a few PSs in the PS-InSAR result. Again, it explained that the fewer Sentinel-1 images affected the reliability of the PS-InSAR results. Thus, the result of the optimal SBAS-InSAR is shown in Figure 17 with the GNSS in the LOS direction. The optimal values indicate significant improvement compared to the SBAS-InSAR values of Xu et al. (2023) [33].

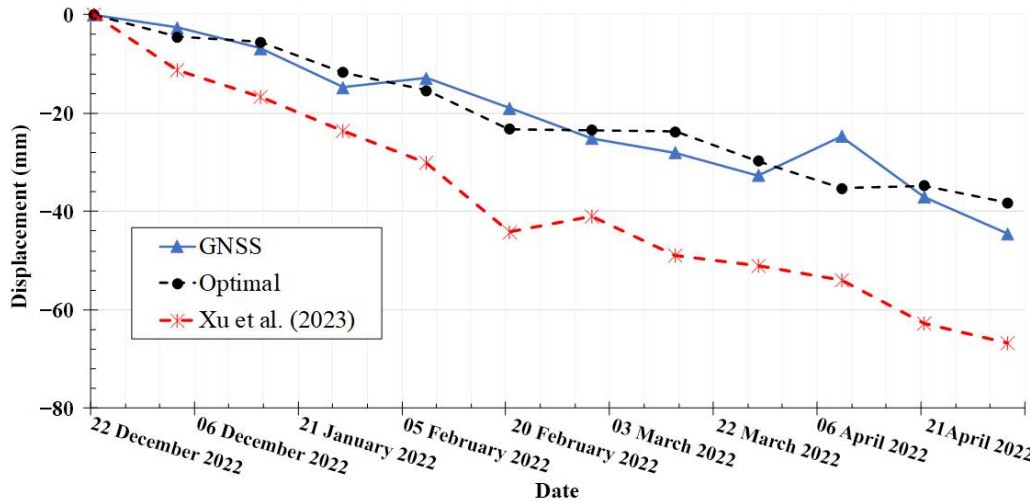

**Figure 17.** The comparison of the optimal SBAS and Xu et al. (2023) [33] results with GNSS.

### 5. Discussion

(1) The parameter shown in Table 1 was tested in the Woda and Shadong landslides. Based on these tests, the significant change between default and optimal parameters was sound. However, the DA depends on the number of InSAR images, and the proper DA threshold proposed in this manuscript is only suitable for the limited number of images at WuWanZai, as well as the Woda and Shadong cases.

(2)  Due to the limitations of landslide monitoring, sometimes only one GNSS station can proceed as the actual displacement reference, such as in the case of WuWanZai and previous studies. Thus, this study aimed to rapidly evaluate the optimal values for this kind of practical landslide implementation using the RMSE compared to GNSS positioning, not the value of velocity standard deviation from the InSAR. However, during the testing and validation, the lowest value of the RMSE from GNSS could not be closer to zero. This was probably caused by the shift from the wrapped phase to the relatively unwrapped phase [48]. Initially, the StaMPS method provided a solution to this problem by estimating the value in the unwrapping step concerning the reference PS in each time increment, leading to the three-dimensional unwrapping [37].

(3)  The Woda and Shadong results indicated only a few PSs in the PS-InSAR result, resulting in the difficulty of GNSS comparison. This explained that the fewer Sentinel-1 images affected the reliability of the PS-InSAR results. Fortunately, the images of the SBAS-InSAR were sufficient for the analyses in both cases. The SBAS-InSAR results were much closer to the GNSS positioning with the optimal suggestions.

(4)  In the PS-InSAR processing, the atmospheric phase screen (APS) plays a pivotal role in accurately estimating deformation values. The spatial–temporal variations due to APS are the dominant error source in interferograms as ghost fringes unrelated to topography or deformation [55]. Although this study did not consider the APS effect for accurate landslide monitoring, the significance of APS should be added in the next phase for comprehending its contribution.

(5)  The early warning threshold for landslides is challenging. The current Sentinel 1A sampling rate for specific areas is over ten days, resulting in limited temporal resolution and insufficient real-time early warning for landslides during rainfall or earthquake events. Thus, fusion analysis of multi-resolution SAR images from different satellites is preferred [56], as well as the enhancement from multi-sensor and multi-scale approaches [57].

## 6. Conclusions

Analyzing the slope movement with the Interferogram Synthetic Aperture Radar (InSAR) measurement is efficient since many traditional monitoring systems are limited in spatial coverage. However, the limitation of the InSAR data processing was a great challenge to be solved in order to obtain a valid measurement result. Therefore, to answer the challenge, this study discussed several influential parameters in Persistent Scatterer (PS)- and Small Baseline Subset (SBAS)- InSAR.

Nevertheless, due to its radar wavelength, the Sentinel-1A, an open-source SAR satellite, could not be applied comprehensively within high vegetation areas. Based on this study, the authors found that the PS- and SBAS-InSAR techniques combined with the open area and typical satellite imagery can generate an assessment of landslide issues, especially for vegetation areas. Influential parameters based on each optimal range value were found for the StaMPS processing, such as $0.47 \leq$ amplitude dispersion $\leq 0.48$, unwrap_grid_size $= 20$, unwrap_gold_alpha $= 0.8$, unwrap_gold_n_win $\leq 32$, and unwrap_time_n_win $\leq 100$ for PS, and $\geq 0.6$, 20 m, 0.8, $\leq 24$, $\leq 32$, respectively, for SBAS. Then, these measurements were utilized and found to be appropriate in the WuWanZai test. In addition, based on the Woda and Shadong validation cases, the SBAS-InSAR is more feasible than the PS-InSAR in landslide monitoring. However, it is necessary to obtain further comparisons in other locations with various environments to verify the accuracy of the proposed optimal parameters.

**Author Contributions:** Conceptualization, F.N.B. and C.-C.C.; methodology Methodology, F.N.B. and C.-C.C.; software, F.N.B.; validation, F.N.B. and C.-C.L.; formal analysis, F.N.B.; investigation, F.N.B.; data curation, F.N.B. and C.-C.L.; writing—original draft preparation, F.N.B.; writing—review and editing, C.-C.C.; visualization, F.N.B. and C.-C.L.; supervision, C.-C.C.; funding acquisition, C.-C.C. All authors have read and agreed to the published version of the manuscript.

**Funding:** This research was supported by the Ministry of Science and Technology (MOST), R.O.C. under Grant 109-2625-M-008-003.

**Data Availability Statement:** The experimental data supporting this work are publicly available online ("http://doi.org/10.5281/zenodo.6805513" (accessed on 7 July 2022)).

**Conflicts of Interest:** The authors declare no conflict of interest.

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
