# Peer review of "Parametric Test of the Sentinel 1A Persistent Scatterer- and Small Baseline Subset-Interferogram Synthetic Aperture Radar Processing Using the Stanford Method for Persistent Scatterers for Practical Landslide Monitoring"

_remotesensing, doi:10.3390/rs15194662_

Round 1
Reviewer 1 Report
For this manuscript, it introduces a parametric test of the sentinel 1A PS- and SBAS-InSAR processing using staMPS for practical landslide monitoring. It need minor revision, and some shortcomings need to be improved.
1. Figure 1 is not clear, need to improve image quality and remove Chinese characters from the image.
2. Can you explain how to determine the three sliding surfaces in Figure 2 ?
3. The format of the formula is incorrect and needs to be edited again
4. Are there actual landslide monitoring results compared with the monitoring results?
The English quality basically meets the requirements for article publication.
Author Response
please find the responces as attachment

Reviewer 2 Report
The manuscript titled "Parametric Test of the Sentinel 1A PS- and SBAS-InSAR Processing using StaMPS for Practical Landslide Monitoring" deals with the critical issue of parametric testing for Multi-temporal InSAR products for landslide monitoring. Indeed the topic dealt with here is very interesting.
Research Questions are very clearly identified.
There is a scarcity of mention of recent studies in the Literature Review. Authors can refer to :
https://link.springer.com/article/10.1007/s10346-023-02097-5
https://www.sciencedirect.com/science/article/pii/S0926985122002257
In Figure 13 figure needs to be improved.
Results and Discussion are very clear and precise.
There are a few English errors such as "while kinds of the literature" in para under 4.1.1.
There are few grammatical errrors
Author Response
please find the responses as attachment

Reviewer 3 Report
Dear authors, thank you for the manuscript. It regards comparison of two InSAR methods, with several parameter values, with GNSS results on three test localities.
Unfortunately, InSAR basics and the basic difference between the two methods tested are not clearly described in the manuscript, confusing the reader even more. There are lots of inproperly used terms, such as spectral / spatial filtering etc.
Paper purpose - testing parameters - does not bring much use to the InSAR community. In case of DA threshold, the software-default values are confirmed - and the results are tested only with regard to GNSS, but without regard to the quality of PS/SBAS points (displacement velocity standard deviation). In case of uwrapping time window, it is evident that it should be smaller than the length of the dataset, which is not the case in the case studies. Regarding "unwrap grid size", I don't understand why the optimal range is 10-50 m, if the Sentinel-1 resolution is 5x20 m (is it really in meters?), especially in case of spatially filtered SBAS interferograms.
Due to ambiguous principle of InSAR, RMSE values (are they in mm??) of 10 mm or more are probably a pure noise (for Sentinel-1; the thresholds is around 5-6 mm). The reliability of InSAR results with regard to the ambiguities shall be also taken into account - to be able to distinguish the cases of mapping pure noise.
It is generally known that mapping landslides in vegetated areas using InSAR is a challenging task.
The text should be made clearer.
Author Response
please find the responses as attachment

Reviewer 4 Report
This evaluation offers feedback on the manuscript titled "Parametric Test of the Sentinel 1A PS- and SBAS-InSAR Processing using StaMPS for Practical Landslide Monitoring." The paper delves into a captivating subject with promising implications for future readers. Following a meticulous assessment of the manuscript, I would like to present the following observations:
1. It has come to my attention that there are some formatting and production errors that should be rectified as a matter of priority.
2. Its is not clear how many datasets are used; I counted the PS-InSAR data point in the graph and figure 8 and found that total 16 images area used to cover the period of 8 months. Author shall add the baseline graph of the images used.
3. PS-InSAR and SBAS technique need more images in order to achieve the reliable results. For SBAS there any many references suggesting the use of at least 20 images to generate the good results. Using mere 16 images, I really doubt the outcome.
4. The overall composition of the paper lacks coherence; certain sections fail to comprehensively address the necessary details. It is recommended to provide a more comprehensive insight into GNSS, PS-InSAR and SBAS methodologies.
5. The author utilizes 16 Sentinel-1 images spanning 8 months. However, there is a wealth of additional data available that could potentially yield more robust outcomes. Incorporating the latest dataset could potentially enhance the research's credibility.
6. The author should provide a clear rationale for why SBAS and PS-InSAR are used and what is the difference. There should be comparison of the result of PS-InSAR, SBAS and GNSS. A comparative analysis of the results would be invaluable for future researchers and readers seeking to understand the context.
7. The study exclusively employs single-polarization VV data, even though dual/quad-polarization data are more pertinent for vegetation-related analyses. It would be beneficial to elaborate on how InSAR results vary with different polarizations, particularly when VV polarization is chosen.
8. In the PS-InSAR processing, the Atmospheric Phase Screen (APS) plays a pivotal role in accurately estimating deformation values. Additional elucidation regarding the significance of APS would aid readers in comprehending its contribution.
9. To enhance clarity, further details could be provided on which DEM was employed for correction. Additionally, any consideration or experimentation with higher-resolution DEMs could be highlighted.
I believe that taking these suggestions constructively into account will substantially enhance the quality of this work for a broader audience's benefit. Thank you
Author Response
please find the responses as attachment

Round 2
Reviewer 3 Report
Dear authors, thank you for the revised manuscript. I have to admit that the language quality has been improved and the manuscript is more comprehensible, but I still have the problem with the methodology: InSAR sometimes gives results that have nothing in common with reality, that represent a pure noise. With high vegetation, if you increase the D_A threshold and phase noise is higher than 5.5 mm, I am afraid that this may be the case. And in case of just one GNSS station, you have no possibility to recognize this case. As a quality criteria, why don't you consider at least the value of velocity standard deviation, which is the output of StaMPS? I hope there are even more quality measures in StaMPS, maybe different for PS and SBAS.
In addition, I could not find any mention of spatial referencing in the manuscript. InSAR is known to give results relative both in space and time, and StaMPS is known to reference in space with regard to the average of all pixels processed. Therefore, if processing only a small area of which most is moving (or the movements are strong), it is expected to get biased results which have to be spatially referenced with regard to a point/area known to be stable. This applies for both PS and SBAS techniques.
If you admit that the number of images is low for PS, resulting in irreliable results, the "parameter optimalization" experiment is senseless. Moreover, the amplitude dispersion index is dependent on the number of images, and therefore the D_A threshold should also be dependent on the number of images. This dependency is very strong for such a low number of images. You tried to find the proper D_A threshold, but only for the not-recommended number of images. Generally, if presenting irreliable results, it is not sufficient to mention it, but rather edit the whole manuscript and exclude them.
It is not explained how RMSE values were calculated (and in the figures, units are missing). This is critical to judge the reliability of the InSAR results. As I disclose above, the phase noise of 5.5 mm and more is the irreliability threshold, and your RMSE values are higher.
The only fact that supports the fact that the InSAR results are reliable, are figures 16 and 17. Please note the unwrapping errors which are common in the reference solutions (differences between GNSS and InSAR close to 28 mm or even more for some image dates).
less serious problems (easy to correct):
- line 53: technology is improved -> HAS improved
- line 62: is separated -> can be divided
- line 65: L-band does not have low spatial resolution, but low accuracy
- on many places: not Permanent Scatter, but Scatterer
- line 161: descending photos (terminology)
- section 3.1 needs to be completely rewritten: difference / similarity of PS/SBAS: the methods may be similar from the user / processing view, but not by principle. I suggest to list their differences clearly rather than say they are similar and describe this in text. "Spatial filtering was not applied in PS-InSAR" - this is a principal difference between the two methods!
- line 201: "conventional SAR method" does not exist, use rather "conventional DInSAR method"
- figure 8: "small baseline interferometric", "interferogram formation and differential interferogram" etc., not comprehensible
- section 3.4.1: amplitude dispersion is calculated for all pixels and this value is thresholded -> pixel selection. Unclear explanation
- "wherein the PS-InSAR method will respect a master image": I don't understand this at all
- section 3.4.2. should be called "phase unwrapping" ?
- lines 251-252: "the iterative process.... and the displacement term" -> "the displacement signal obstructions are evaluated in an iterative process"?
- line 260: "non-spatial correlation" senseless. rather "spatially uncorrelated"?
- line 274: compared -> comparing
- line 311: discuss the unwrap grid size in pixels (in addition to meters)
- chapter 4: please include the procedure to calculate RMSE and add units to figures
- section 4: it would be very useful if you added a description of how to distinguish reliable results from a pure noise
- line 335: the last sense does not make sense
- line 355, 357: scatter -> scatterer
The Discussion section is still incomprehensible and needs to be rewritten.
Reviewer 4 Report
Dear Authors,
thanks for the responses. But question 5 is not implemented. I was expecting it will be included.
